# A Humanized Bone Niche Model Reveals Bone Tissue Preservation Upon Targeting Mitochondrial Complex I in Pseudo-Orthotopic Osteosarcoma

**DOI:** 10.3390/jcm8122184

**Published:** 2019-12-11

**Authors:** Ivana Kurelac, Ander Abarrategi, Moira Ragazzi, Luisa Iommarini, Nikkitha Umesh Ganesh, Thomas Snoeks, Dominique Bonnet, Anna Maria Porcelli, Ilaria Malanchi, Giuseppe Gasparre

**Affiliations:** 1Dipartimento di Scienze Mediche e Chirurgiche, Università di Bologna, Via Massarenti 9, 40138 Bologna, Italy; nikkitha.umesh@gmail.com (N.U.G.); giuseppe.gasparre3@unibo.it (G.G.); 2Tumor-Host Interaction Lab, The Francis Crick Institute, 1 Midland Rd, London NW1 1AT, UK; ilaria.malanchi@crick.ac.uk; 3Hematopoietic Stem Cell Laboratory, The Francis Crick Institute, 1 Midland Road, London NW1 1AT, UKdominique.bonnet@crick.ac.uk (D.B.); 4Regenerative Medicine Lab, CICbiomaGUNE, Paseo Miramón 182, 20014 Donostia, Spain; 5Ikerbasque, Basque Foundation of Science, Maria Diaz de Haro 3, 48013 Bilbao, Spain; 6Anatomia Patologica, Azienda Unità Sanitaria Locale–IRCCS di Reggio Emilia, Viale Risorgimento 80, 42123 Reggio Emilia, Italy; moira.ragazzi@ausl.re.it; 7Dipartimento di Farmacia e Biotecnologie, Università di Bologna, Via Selmi 3, 40126 Bologna, Italy; iommarini.luisa@gmail.com (L.I.); annamaria.porcelli@unibo.it (A.M.P.); 8In Vivo Imaging Operations, The Francis Crick Institute, 1 Midland Road, London NW1 1AT, UK; thomas.snoeks@crick.ac.uk; 9Centro Interdipartimentale di Ricerca Industriale Scienze della Vita e Tecnologie per la Salute, Università di Bologna, Via Tolara di Sopra 41/E, 40064 Ozzano dell’Emilia, Italy; 10Centro di Ricerca Biomedica Applicata (CRBA), Università di Bologna, Via Massarenti 9, 40138 Bologna, Italy

**Keywords:** mitochondrial complex I, osteosarcoma, orthotopic models, tumor microenvironment

## Abstract

A cogent issue in cancer research is how to account for the effects of tumor microenvironment (TME) on the response to therapy, warranting the need to adopt adequate in vitro and in vivo models. This is particularly relevant in the development of strategies targeting cancer metabolism, as they will inevitably have systemic effects. For example, inhibition of mitochondrial complex I (CI), despite showing promising results as an anticancer approach, triggers TME-mediated survival mechanisms in subcutaneous osteosarcoma xenografts, a response that may vary according to whether the tumors are induced via subcutaneous injection or by intrabone orthotopic transplantation. Thus, with the aim to characterize the TME of CI-deficient tumors in a model that more faithfully represents osteosarcoma development, we set up a humanized bone niche ectopic graft. A prominent involvement of TME was revealed in CI-deficient tumors, characterized by the abundance of cancer associated fibroblasts, tumor associated macrophages and preservation of osteocytes and osteoblasts in the mineralized bone matrix. The pseudo-orthotopic approach allowed investigation of osteosarcoma progression in a bone-like microenvironment setting, without being invasive as the intrabone cell transplantation. Additionally, establishing osteosarcomas in a humanized bone niche model identified a peculiar association between targeting CI and bone tissue preservation.

## 1. Introduction

Developing treatments that target cancer metabolic reprogramming is gaining momentum, but it is often neglected that these approaches are not specific only to proliferating cancer cells. Metabolic therapies inevitably exert a systemic effect, influencing also the non-neoplastic cells, including tumor microenvironment (TME), which in turn may completely redirect the final outcome of the disease. Indeed, emerging literature warns that cancer cell metabolism may have crucial consequences on the phenotype of different non-malignant cell types within the tumor [1]. For example, tumor cells may engage in metabolic strategies such as nutrient competition to avoid immune cytotoxicity [2]. On the other hand, targeting cancer metabolic reprogramming may trigger compensatory responses leading to TME-related resistance mechanisms [3]. Thus, strategies directed against cancer metabolism need to be tested in models that allow a proper assessment of the TME effect on tumor progression.

The bone microenvironment has recently been recognized as essential in determining the fate of osteosarcoma development [4,5], even to the extent that targeting TME has been suggested as an efficient strategy to fight the disease [6,7]. In this context, it is interesting to note that studies evaluating the efficacy of antimetabolic therapies in osteosarcoma, such as those against mitochondrial complex I (CI), suggest the TME may play a role in defining the response to treatment [3,8,9]. In line with these findings, we have recently reported that targeting CI arrests progression of osteosarcomas, converting them into low-proliferative, oncocytoma-like lesions, and demonstrated that the loss of hypoxia inducible factor 1-alpha (HIF-1α) is accountable for the antitumorigenic effects of CI dysfunction [3]. However, the latter was associated with an abundance of tumor associated macrophages (TAMs), whose depletion improved the anti-cancer efficacy of metformin, a known CI inhibitor [3]. Thus, even though targeting CI is being recognized as a valid anti-cancer strategy associated with various antitumorigenic effects [10,11,12,13,14,15], at the same time CI inhibitors seem to elicit conflicting consequences on TME which alter the therapy response [16]. 

In murine xenografts, TME is usually taken into account by an orthotopic implant, which consents cancer cell proliferation within the native environment. For some tumor types, this approach is relatively simple, as in the case of injecting breast cancer cells in the mouse mammary fat-pad [17], or in hepatocellular carcinoma setting, where orthotopic tumor models involve a minor surgery [18]. The establishment of in situ osteosarcomas is particularly challenging, as the surgical procedures to reach the tibia or femur are invasive, and the injection is difficult due to bone stiffness, requiring drilling bone plateau and potentially resulting in the leakage of cancer cells [19,20]. Engineering approaches today may be used to create ectopic grafts resembling bone tissue environment [5], allowing not only tumor progression in the native tissue, but also generation of human TME, which is important in the setting where human cancer cells are being investigated [21]. Such approaches are becoming crucial for appropriate investigation of osteosarcoma development, since the bone tumor fate was shown to be influenced by the inoculation environment [5].

Here we take advantage of new methods to humanize and modulate ectopic osteosarcoma graft, with the aim to understand whether CI-deficiency induces changes in bone specific non-neoplastic cells during cancer progression. This pseudo-orthotopic approach established that abundance of cancer associated fibroblasts (CAFs) and TAMs is a hallmark of CI-deficient TME, and identified a peculiar association between targeting CI and osteocyte/osteoblast preservation.

## 2. Materials and Methods

### 2.1. Cell Lines

Osteosarcoma 143B Tk^−^ cells were purchased (#CRL-8303, ATCC, LGS Standards, Milan, Italy) and cultured at low passages (<50) in in Dulbecco’s modified Eagle medium (DMEM) High Glucose (#ECM0749L, Euroclone, Milan, Italy), supplemented with 10% FBS (#ECS0180L, Euroclone), L-glutamine (2 mM, #ECB3000D, Euroclone), penicillin/streptomycin (1x, #ECB3001D, Euroclone) and uridine (50 µg/mL, #U3003, Sigma-Aldrich, Milan, Italy), in an incubator with a humidified atmosphere at 5% CO_2_ and 37 °C. The cell origin was authenticated using AMPFISTRIdentifiler kit (#4322288, Applied Biosystems, Monza, Italy) and their STR profile corresponded to their putative background. Genome editing for generation of NDUFS3 knock-out was performed using zinc finger endonucleases purchased from Sigma-Aldrich (#CKOZFND15168, Milan, Italy), according to the manufacturer’s instructions. 

Primary human mesenchymal stroma cells (hMSCs) were purchased (#PT-2501, Lonza, Slough, UK) and grown in alpha Minimun Essential Medium (αMEM) (#32571-028, Gibco, Paisley, UK), hMSC-specific FBS (10%) (#12662-029, Gibco, Paisley, UK), penicillin/streptomycin (1x, #ECB3001D, Euroclone), and used at low passages (<5).

### 2.2. Establishing Pseudo-Orthotopic Osteosarcomas 

Most steps were performed as previously described [22,23,24]. All pre-surgical procedures were performed in sterile conditions. Gelfoam gelatin sponges (2 cm × 6 cm × 7 mm) (Pfizer, Kalamazoo, MI, USA) were sectioned into 24 pieces, washed with ethanol 70%, rehydrated in sterile PBS and placed in a 24-well plate. hMSC cells (5 × 10^5^ in 50 µl) were injected with a syringe (29G) and left to attach for 4 hours in a 37 °C incubator. Culture media was added and the cells were left to grow for 7 days. On day 8 osteosarcoma cells (10^5^ in 30 µl) were injected into the scaffolds and left to attach before adding fresh culture media. On day 9 the scaffolds were clothed following previously described protocol [22]. Each scaffold was allocated in a 15 mL tube and 8 μL of bone morphogenic protein 2 (BMP-2) (Noricum, Tres Cantos, Spain) (reconstituted in acetic acid 50 mM at 5 μg/μL) were added. Then, 30 μL of thrombin from human plasma (Sigma, Dorset, UK) (reconstituted in 2% CaCl_2_ at 20 U/mL) and 20 μL of fibrinogen from human plasma (Sigma) (PBS reconstituted at 4 mg/100 mL) were incorporated. Solidification was allowed during 30 min in cell culture conditions before proceeding with in vivo implantation.

Surgery was performed in aseptic conditions. Five to six-week old female *Rag1*^−/-^
*FVB/n* mice available at The Francis Crick Institute Biological Research Facility (London, UK) were used. The animals were treated according to institutional guidelines and regulations and experiments performed in accordance with UK Home Office regulations under project license PPL number P83B37B3C. A bilateral implantation was performed. In detail, 2 hours before surgical procedure caprofren (Rimadyl, Zoetis, Leatherhead, UK) anti-inflammatory and pain-killer drug was administrated to each animal, both subcutaneously and in the drinking water. Anesthesia was induced with 2.5% isoflurane and O2 at 2–4%. A wide section of fur from the back was shaved. Then skin was sterilized twice with surgiscrub. For each scaffold implantation, 0.5 cm vertical incision was made 1 cm away from the spine on each side of the animal. With forceps, a pocket under the skin was made in the incision, down the side of the animal. A scaffold was inserted, making sure it was placed deep within the pocket, and then incisions were dried and glued (3M surgical glue, Vetbond, St Paul, MN, USA). Buprenorphine (Vetergesic, Alstoe, York, UK) post-operative analgesia was administrated subcutaneously. Animals were placed in a pre-warmed cage and left to recover. After surgery, animals were checked frequently for their well-being. Rimadyl in the drinking water was removed 48 hours after surgery. Mice were sacrificed either at 30 or at 60 days post implantation.

### 2.3. Micro Computed Tomography Imaging 

Samples were scanned using a SkyScan-1176 μCT scanner (Bruker MicroCT, Kontich, Belgium). The X-ray source was operated at 40 kV and 600 μA, no filter was used. The scans were made over a trajectory of 180° with a 0.5° step size with a 8.57μm pixel size. The images were reconstructed using nRecon (Bruker MicroCT, Kontich, Belgium) and further analysed using CTan (Bruker MicroCT).

### 2.4. Histology

Tumor tissue was processed following standard immunohistochemistry protocols. Before embedding, the samples were decalcified with 17% EDTA (Osteosoft, #101728, Merck Millipore, Watford, UK) for 7 days. Hematoxylin/eosin coloration was performed following standard protocol and collagen fibers staining with the Masson’s Trichrome Stain Kit (#25088, Polysciences, Hirschberg an der Bergstrasse, Germany). The following primary antibodies were used: mouse monoclonal anti-HIF-1α (1:100, #610959, BD Biosciences, Berkshire, UK); mouse monoclonal anti-KI-67 (1:100, #M7240, Dako, Agilent, Cernusco sul Naviglio, Italy); rat anti-endomucin (1:200, #SC-65495, Santa Cruz, DBA, Segrate, Italy); mouse anti-SMA (1:750, #M0851, Dako) and rat monoclonal F4/80 (1:100, #14-4801, eBiosciences, ThermoFisher, Life Technologies, Monza, Italy). For evaluation of KI-67 positive nuclei, only cancer cells were counted at 60× magnification in one hot spot area per tumor, avoiding stromal infiltrations and necrotic tissue. Macrophages (F4/80+) were counted at of 20× magnification in three fields of view (FOV) per tumor. The macrophages located close to trabecular bone were counted by considering F4/80 positive cells touching the bone matrix. The macrophages infiltrating the tumor tissue were counted by avoiding tumor front, trabecular bone and necrotic tissue. Osteocytes and osteoblasts were counted in three consecutive FOV at 60× magnification, in proximity to the trabecular bone, starting from the hot spot area.

Immunofluorescent staining included 15 min citrate antigen retrieval (10 mM sodium citrate, pH = 6) at 95 °C, 10 min blocking with goat serum (#156046, Abcam, Cambridge, UK) at RT, 1 hour incubation with primary antibodies at RT (rat anti-endomucin (1:200, #SC-65495, Santa Cruz) and mouse anti-SMA (1:750, #M0851, Dako), 40 min incubation with Alexa Fluor (ThermoFisher, Life Technologies, Monza, Italy) secondary antibodies at RT (488-goat anti-mouse diluted 1:500 and 555-goat anti-rat diluted 1:350) and mounting with Vectashield Antifade Mounting Medium containing DAPI (#H-1200, Vector Laboratories, Peterborough, UK). Vessel size was evaluated by measuring the longer diameter of 20 endomucin positive cells per tumor and avoiding areas of collective fibroblast infiltration. Immature vessels (Endo+SMA−) were counted in five FOV at 20× magnification per tumor.

### 2.5. Flow Cytometry

Xenograft samples (approximately 50 mm^3^) were digested immediately after the sacrifice for 40 minutes at 37 °C with Liberase TL (#5401020001, Sigma), Liberase TM (#5401135001, Sigma) and DNaseI (#DN25, Sigma) in HBSS and passed through a 100 µm strainer. Hypotonic lysis with Red Blood Cell Lysis Buffer (#11814389001, Sigma) was performed and remaining cells were washed with MACS buffer (2 mM EDTA, 0.5% BSA in PBS), blocked using FcR Blocking Reagent (#130-092-575, Miltenyi, Surrey, UK) and incubated with panels of pre-labelled antibodies. In parallel, spleen, lung and a control tumor tissue were digested together and stained for Fluorescence Minus One (FMO) reading which was considered while setting the gating strategy. The following panels were used: Panel 1 (for analysis of the tumor macrophage, neutrophil and dendritic cell contribution): anti anti-CD45-APC (clone 30-F11, #17-0451-82, eBioscience), anti-CD11b-ef450 (clone M1/70, #48-0112-82, eBioscience), anti-F4/80-FITC (clone BM8, #123108, BioLegend, London, UK), anti-Ly6G-APC780 (clone RB6-8C5, #47-5931-80, eBioscience), anti-CD11c-PE (clone n418, #12-0114-81, eBioscience); Panel 3 (for analysis of M1/M2 protumorigenic macrophages): anti-CD45-APC780 (clone 30-F11, #47-0451-80, eBioscience), anti-F4/80-ef450 (clone BM8, #48-4801-82, eBioscience), anti-CD206-APC (clone C068C2, #141707, BioLegend). All antibodies were used at 1:100 dilution, apart from the anti-CD45 which was diluted 1:300. Between 300,000-500,000 cells were stained. Dead cells were stained with DAPI and gated out for analyses. Absolute cell abundance was defined as their percentage among all live cells (%Live). Relative cell abundance was defined as their percentage among populations indicated in the figure panels. The samples were run on LSRFortessa cell analyzer (BD Biosciences) and data was analyzed by BD FACSDIVA Software (BD Bioscience) and Flow Jo (Tree Star Inc., Ashland, OR, USA) software.

### 2.6. Cytokine Profiling

Xenograft-derived cell cultures were generated by a 10-day cultivation of liberase-digested tissue in basal conditions. Supernatant (0.5 mL) was taken 2 days after medium renewal from a 500,000 cells cultured and analyzed with human Proteome Profiler Array kit (ARY005B, R&D Systems, Abingdon, UK) following manufacturer’s instructions. ImageJ was used for quantification of the dot blots.

### 2.7. Statistical Analyses

GraphPad Prism version 7 (GraphPad Software Inc., San Diego, CA, USA) was used to perform statistical tests and create bar plots and graphs. Unless stated otherwise, a two-tailed unpaired Student’s t-tests assuming equal variance were performed to compare averages. For each experiment, *p*-values (* *p* < 0.05, ** *p* < 0.01, *** *p* < 0.001) are indicated in the graphs. 

## 3. Results 

### 3.1. A Humanized Bone Niche Scaffold Recapitulates Mature Bone Characteristics and Serves as a Pseudo-Orthotopic Osteosarcoma Xenograft Model

A humanized bone-forming ectopic xenotransplantation model was set up by using hMSCs treated with BMP-2, and applied as pseudo-orthotopic approach to grow CI-competent (143B^+/+^) and CI-deficient (143B^−/−^) osteosarcoma cells in vivo (Figure 1a). 

Targeting CI reduced tumorigenic potential of 143B cells at day 30, but this antitumorigenic effect was less appreciated by day 60, when tumor size and Ki-67 proliferation index were similar between the two groups (Figure 1b,c). 

Morphologically, the control sample, in which no tumor cells were injected, displayed mature spongy bone consisting of osseous trabeculae surrounded by adipose cells, blood vessels and leukocyte infiltration encompassing polymorphonuclear neutrophils (Figure 2a,b). Furthermore, lamellar bone matrix contained lacunae with typical bone-specific cell populations such as osteocytes and a thin layer of osteoblasts lined bony spicules (Figure 2b). Of note, this model allows effective humanized bone microenvironment formation [23,24]. 

The masses deriving from scaffolds seeded with 143B cells displayed trabecular bone containing osteocytes and osteoblasts, as well as occasional adipose tissue areas, but were primarily occupied by neoplastic cells, indicating their high degree of aggressiveness (Figure 2c). Micro computed tomography (Micro CT) imaging identified calcified areas in the xenografts similar to native bone parenchyma, indicating the formation of a bone-like tissue (Appendix A). These observations confirmed the establishment of a mature bone tissue in which human osteosarcoma cells may progress within a bone-like microenvironment setting.

### 3.2. Tumor Associated Macrophages are the Main Hallmark of the CI-Deficient Osteosarcoma Microenvironment 

With the aim to unravel the potential contribution of TME associated to CI-deficient osteosarcoma, we first carried out a detailed analysis of immune cells within the xenografts. A higher neutrophil count was detected in the control tumors, whereas TAMs were more abundant in CI-deficient masses (Figure 3a,b, Appendix A). No difference in TAM polarization from inflammatory M1 towards protumorigenic M2 population was observed, but the general contribution of M2 macrophages in the mass was higher in tumors lacking CI (Figure 3c). Interestingly, immunohistochemistry revealed TAMs were mainly located on the tumor front in the control masses, whereas in CI-deficient osteosarcomas they were also infiltrating the tumor mass and were often located in the proximity to the trabecular bone (Figure 3d).

We next sought to understand which factors are contributing to the difference in macrophage abundance. Xenograft tissue-derived supernatants were blotted on a human cytokine array and macrophage migration inhibitory factor (MIF) was found downregulated in CI-deficient tumor-derived secretome (Figure 4a). MIF is a HIF1-responsive gene, whose downregulation has been associated with triggering a TAM-mediated alternative proangiogenic activity as a compensatory response upon anti-VEGF treatment [25].

Thus, we analyzed HIF-1α levels and localization in the xenografts, as well as their vascular architecture, since myeloid-derived proangiogenic signals have been associated with small vessels which lack pericyte marker smooth muscle actin (SMA) [26]. Whereas the control tumors expressed HIF-1α in their cancer cell nuclei, 143B*^−/−^* masses were characterized by a complete absence of HIF-1α staining (Figure 4b), in line with MIF downregulation. Interestingly, the lack of HIF-1α in CI-deficient tumors was associated with a higher total number of vessels, but these were significantly smaller than the ones found in controls, and were mostly lacking lumen and pericyte coating (Figure 4c), a phenotype analogue to the abnormalized vasculature typical of myeloid-derived proangiogenic signals [26].

Overall, these data identify protumorigenic macrophages as the most abundant TME component in 143B^−/−^ xenografts and suggest their recruitment may be a consequence of MIF downregulation. 

### 3.3. Osteocytes and Osteoblasts within Trabecular Bone are Preserved in the CI-Deficient Pseudo-Orthotopic Osteosarcoma Xenografts

To characterize the bone-specific TME during osteosarcoma progression, we next analyzed histology of mesenchymal cell populations in the humanized bone xenografts. A prominent involvement of the stromal cells was evident in the intra-tumoral septae specifically in CI-deficient tumors, as supported by the Masson’s trichrome collagen staining and CAF marker SMA immunohistochemistry (Figure 5a,b). Despite the trabecular bone and osteoid matrix were noticeable in both groups, control xenografts were characterized by intensive necrosis which appears as tissue areas replaced by granulocytes and cellular debris with loss of nuclei. Moreover, at 60 days post implantation, osteonecrosis was evident in controls, characterized by loss of osteocytes with empty osteocytic lacunae in the mineralized bone matrix (Figure 5c). On the other hand, no necrotic tissue was observed in 143B*^−/−^* tumors, which displayed higher number of osteocytes and osteoblasts in the trabecular bone when compared to the controls (Figure 5c). Moreover, consistently with histology, micro CT scans of CI-deficient tumors showed higher mean intensity of the calcified volume (Figure 5d), indicating this condition is associated with lower osteolytic activity and preservation of the bone microenvironment.

## 4. Discussion

In this study, we show that by using a humanized niche model of the osteosarcoma graft, an additional level of information about the tumor histology is achieved with respect to canonical subcutaneous implant. The peculiar bone microenvironment preservation in CI-deficient tumors highlights that parenchymal cells are an important component of TME, warning they should not be neglected when investigating cancer progression. Indeed, functionally relevant cancer associated parenchyma has recently been described also in the setting of breast cancer metastases [27].

The differences regarding bone specific cell types appreciated depending on the condition tested, allow to hypothesize that osteocytes and osteoblasts may influence the response to therapies designed against CI. Targeting CI in osteosarcomas grown in the humanized bone reduced tumorigenic potential of 143B cells, albeit not as strikingly as observed in our previously described experimental settings [3,14,15]. Among other, these milder consequences may be due to the osteocyte/osteoclast-specific functions. Their preservation was particularly associated with the later stages of tumor progression, at which the antitumorigenic effect of targeting CI was less appreciated, suggesting that bone-specific non-neoplastic cells may be involved in promoting osteosarcoma survival. The possible mechanisms of growth support may be relative to essential metabolites exchange between cancer and TME cells, as previously suggested [28,29]. On the other hand, TME may sustain cancer cell proliferation by promoting angiogenesis. In this context, CI deficiency was also associated with the abundance of TAMs, that have been called into play to provide angiogenic factors when cancer cell-autonomous HIF1 signals are absent [25,26]. Indeed, the observation of TAM abundance and vasculature typical of myeloid cell proangiogenic activity in the context of the orthotopic CI-deficient xenografts corroborates our previous findings [3], suggesting that targeting CI in 143B osteosarcoma prevents HIF1-MIF activation, leading to TAM accumulation and vascular architecture remodeling.

Further investigation is required to understand the significance of these data, by using larger animal cohorts and CI inhibitors, rather than the genetic disruption of the complex, since a drug will inevitably act on TME cell populations as well [16]. Moreover, adaptive immunity should be taken into consideration by using immunocompetent models. In this context, it is important to note that a murine bone niche could be easily generated by populating the scaffold with murine MSCs. Interestingly, a study evaluating the effects of CI inhibitor metformin in immunocompetent settings reported reduced number of myeloid derived suppressor cells and TAMs in osteosarcomas [9]. The authors worked with intra-dermal grafts, therefore it would be of interest to understand what is the effect of CI inhibition on bone specific TME cells, such as osteocytes and osteoclasts. This would be particularly important since, in the bone tumor context, the fate of cancer cells in vivo depends on the type of graft that is being used as a model. For example, subcutaneous injection of transformed bone marrow mesenchymal cells is associated to development of leiomysarcoma-like tumors, while the intrabone transplantation of the same cells induced metastatic osteoblastic osteosarcoma, underlining the importance of signals elicited by the bone TME [5]. 

Taken together, preservation of osteocytes and osteoblasts observed upon targeting CI points to the importance of setting up appropriate tumor models, which take into consideration the origin of the cancer in question, since apart from immune cell populations, the parenchymal cells of the TME may also influence neoplastic development.

## Figures and Tables

**Figure 1 jcm-08-02184-f001:**
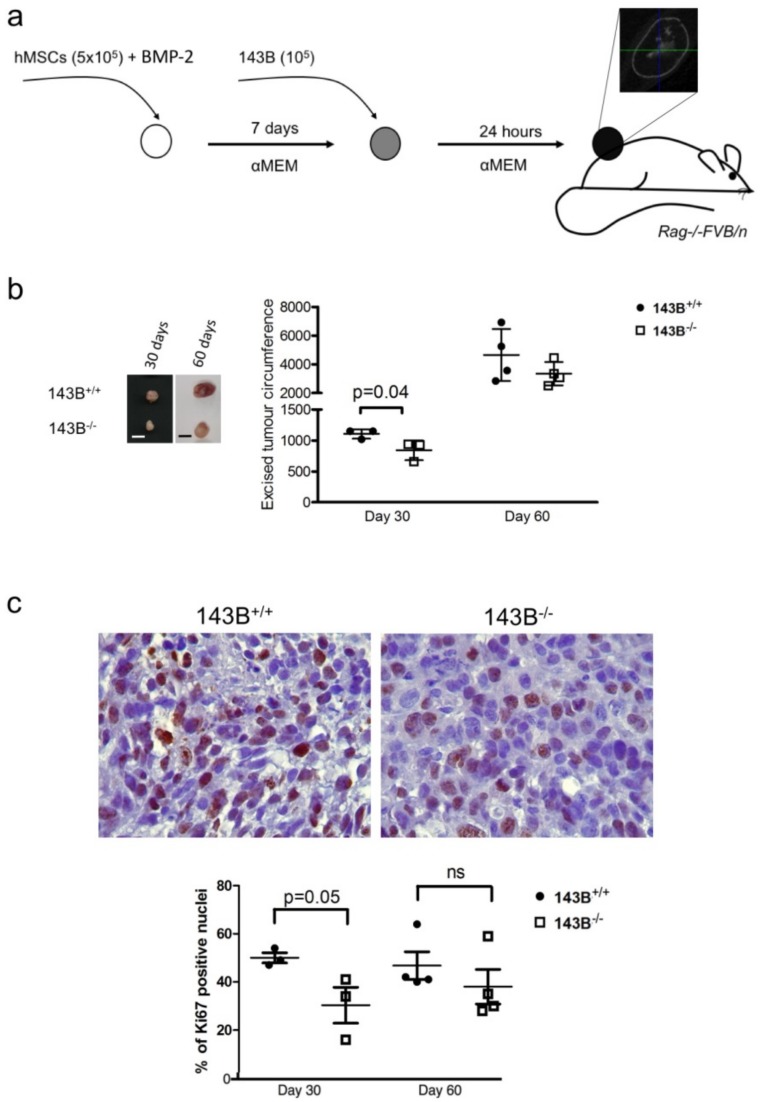
Osteosarcoma progression in the humanized bone niche model. (**a**) Experimental setting: mesenchymal stroma cells (hMSC) and osteosarcoma cells (143B) were seeded in gelfoam scaffolds (circles), treated with bone morphogenic protein (BMP-2), cultured with alpha Minimum Essential Medium (αMEM) and implanted in immunodeficient *Rag*−/−/*FVB/n* mice. A micro CT scan of control sample (hMSC+BMP-2) is displayed. (**b**) Xenograft size of the CI-competent (143B^+/+^) and CI-deficient (143B^−/−^) cells. Representative tumors are shown. Bars = 1 cm. One-tailed T-test was used to calculate statistical significance. (**c**) Quantification of Ki-67 positive nuclei. Representative images are shown for Ki-67 staining in xenografts excised at day 30. Magnification 60×. One-tailed T-test was used to calculate statistical significance.

**Figure 2 jcm-08-02184-f002:**
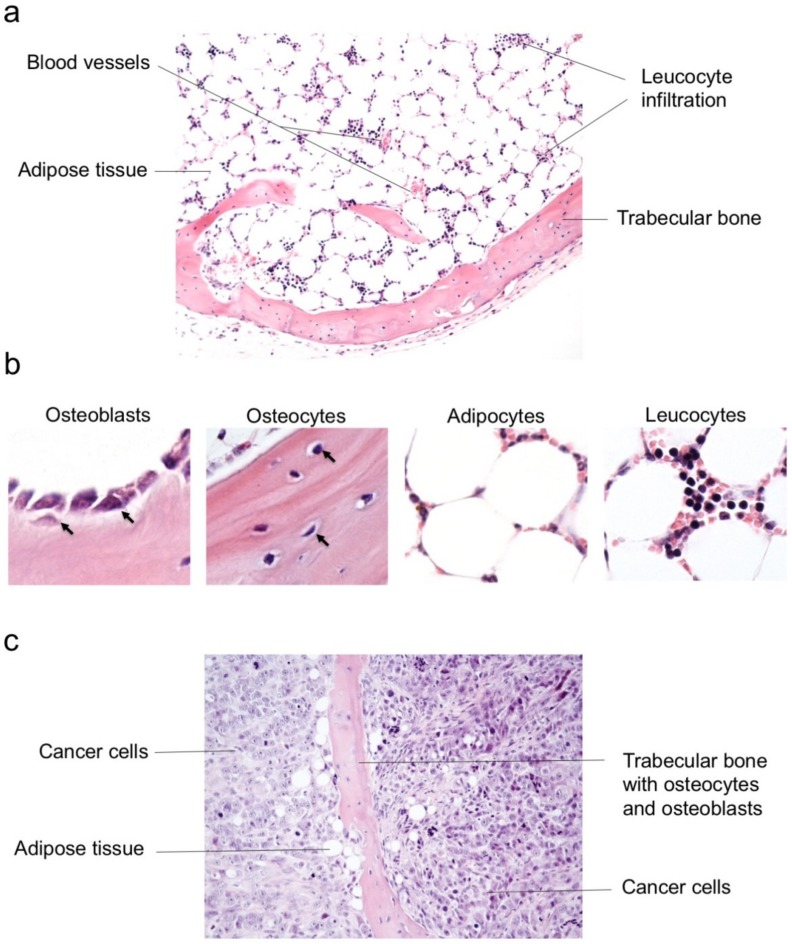
The humanized bone niche model allows investigation of bone-specific cell populations within a growing tumor. (**a**) Hematoxylin and eosin staining showing morphology within the humanized bone niche control sample. Magnification 10×. (**b**) Hematoxylin and eosin staining of the humanized bone niche control sample reveals columnar or flatter osteoblasts covering the bone surface while spindle-shaped and regular osteocytes are embedded in the mineralized bone matrix. Leukocyte infiltration encompassing some granulocytes is observed between adipocytes. Magnification 40×. (**c**) A representative image of hematoxylin and eosin stained CI-deficient osteosarcoma xenograft in which bone specific cell types may be recognized. Magnification 20×.

**Figure 3 jcm-08-02184-f003:**
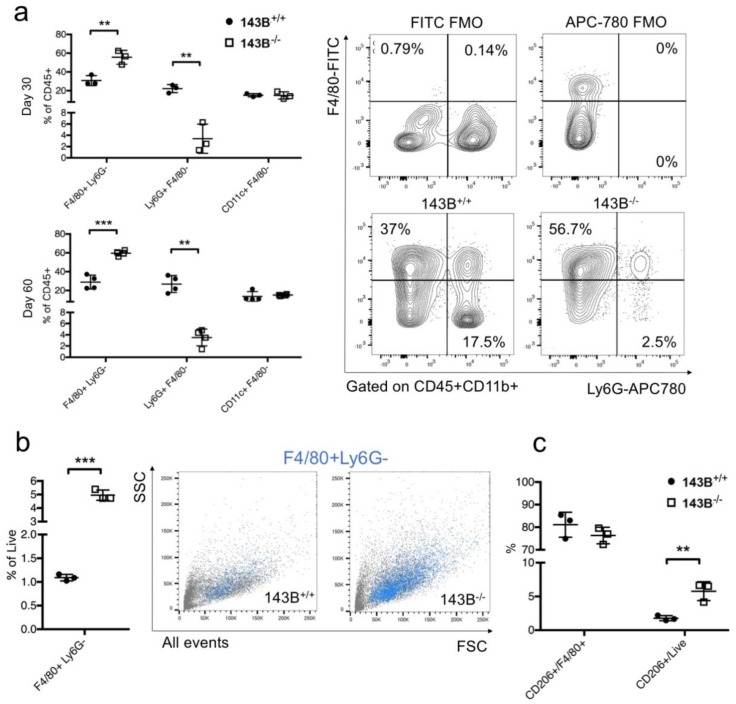
Tumor associated macrophages are a hallmark of CI-deficient osteosarcoma xenografts. (**a**) Flow cytometry analysis of innate immune system populations in 143B xenografts at day 30 (*n* = 3) and day 60 (*n* = 4) post-implantation. The contribution of macrophages (F4/80+Ly6G-), neutrophils (Lys6G+F4/80-) and dendritic cells (CD11c+F4/80-) is shown. Single values are displayed, with the error bars representing standard error of the mean. Representative contour plots with outliers are shown for evaluation of macrophage and neutrophil numbers. FMO: Fluorescence Minus One (**b**) The contribution of macrophages in 143B^+/+^ (black circles) and 143B^−/−^ (white squares) tumors at day 30 as evaluated by flow cytometry. Single values are displayed, with the error bars representing standard error of the mean. Representative dot-plots display contribution of macrophages (blue, F4/80+Ly6G-) among 100,000 acquired events. SSC: Side SCatter; FSC: Forward SCatter (**c**) The relative and absolute contribution of CD206+ macrophages in osteosarcoma tumors at day 30. Single values are displayed, with the error bars representing standard error of the mean. (**d**) Representative images of immunohistochemistry analysis for macrophage marker F4/80 in osteosarcoma xenografts at day 30. Scale bars: 50 µm. Dashed line indicates tumor front. The numbers of macrophages located close to trabecular bone (images in the upper panels) and infiltrating the tissue (images in the lower panels) are graphed. Single values are displayed, with the error bars representing standard error of the mean. FOV: Field of View. In each graph, statistical significance is specified with asterisks (* *p* < 0.05, ** *p* < 0.01, *** *p* < 0.001).

**Figure 4 jcm-08-02184-f004:**
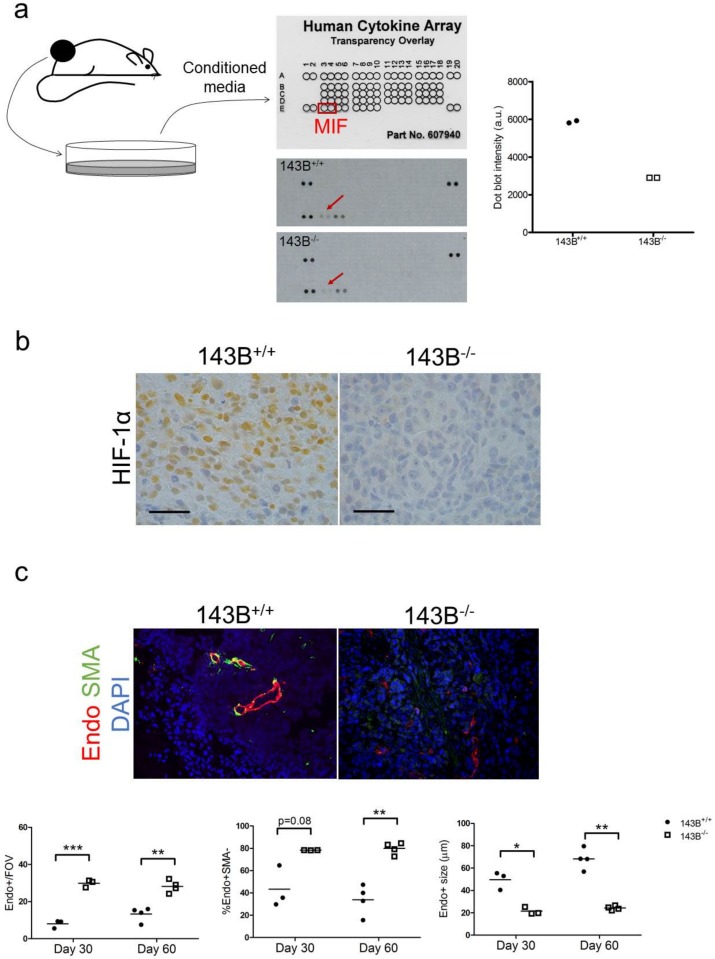
The HIF1-MIF axis is inactive in CI-deficient xenografts. (**a**) Experimental setting and results of the cytokine screening in xenograft-derived cell culture supernatants. The arrows indicate the cytokine array dot blots for MIF. Dot blot pixel intensity for MIF is graphed. (**b**) Representative images of immunohistochemistry staining for HIF-1α in osteosarcoma xenografts. Scale bars: 50 µm. (**c**) Representative images of immunofluorescent staining analyzing vessel morphology in osteosarcoma xenografts. Endo – Endomucin. Endothelial cells (Endo+), pericytes (SMA+Endo+), CAF (SMA+Endo-), nuclei (DAPI). Magnification 20×. Graphs show total number of vessels per field of view (FOV), percentage of pericyte negative vessels (%Endo+SMA−) and the average vessel sizein CI-competent and CI-deficient 143B tumors. In each graph, statistical significance is specified with asterisks (* *p* < 0.05, ** *p* < 0.01, *** *p* < 0.001).

**Figure 5 jcm-08-02184-f005:**
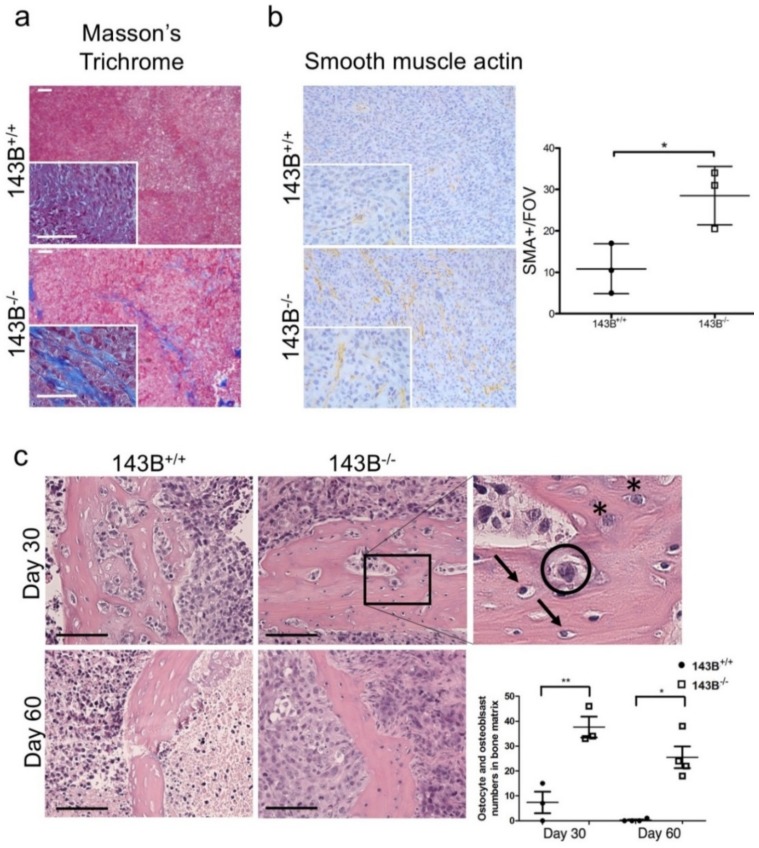
CI-deficient osteosarcomas are associated with preservation of the bone microenvironment. (**a**) Representative images of Masson’s trichrome staining of the osteosarcoma xenografts. Collagen is stained in blue. Scale bars: 100 µm. (**b**) Smooth muscle actin (SMA) immunohistochemistry staining and count of SMA positive cells at 30 days post implantation in CI-competent (143B^+/+^) and -deficient (143B^−/−^) tumors. Magnification 20×, inserts 60×. (**c**) Representative images of hematoxylin and eosin staining of the trabecular bone in the osteosarcoma xenografts at day 30 and day 60 post implantation. The arrows indicate osteocytes, the asterisks osteoblasts, whereas the neoplastic cells are circled. Scale bars: 100 µm. The graph represents the quantification of osteocytes and osteoblasts in the osteosarcoma xenografts. (**d**) Representative micro CT scan images from CI-competent and deficient osteosarcoma tumors. Mean intensity of the calcified volume in the tumors is graphed. In each graph, statistical significance is specified with asterisks (* *p* < 0.05, ** *p* < 0.01).

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
