# Peer review of "A Humanized Bone Niche Model Reveals Bone Tissue Preservation Upon Targeting Mitochondrial Complex I in Pseudo-Orthotopic Osteosarcoma"

_jcm, 2019, doi:10.3390/jcm8122184_

Round 1

Reviewer 1 Report

This study by Kurelac et al. presents an initial characterization of the tumor microenvironment of a humanized bone niche ectopic graft with mitochondrial respiratory complex I-deficient tumors. This report demonstrates the both utility of the utilizing a humanized bone niche model when investigating osteosarcoma and differences in the TME in control versus CI-deficient tumors, and therefore, is a relevant contribution to the osteosarcoma field. Importantly, in general the authors' molecular characterization support their histological observations. Overall, this report is well-organized, clearly written, and the interpretations and conclusions generally fit the data presented. In order to enhance the clarity of their paper and increase the utility of this study to the general readership of the Journal of Clinical Medicine, this reviewer recommends the following minor edits.

1) The basis for this study conducted by the authors is to demonstrate the unitlty of placing particular cancers in their proper so that the TME can be taken into account. However, the background information related to the basis for their study is too short to provide the appropriate context for their study. In the current draft, there is only one short paragraph on the general background of TME (with zero references), another short paragraph on CI-associated tumors and TME (only 4 references for the bulk of the paragraph, and 6 references attributed to one generalized statement), and less than one paragraph on osteosarcoma TME (only 4 osteosarcoma-specific references). Together, this background only provides the reader with less-than-sufficient material to fully appreciate the utility of this study. The authors should significantly extend the breadth of their background, capturing a more representative sampling of the important literature upon which they built their study.

2) The authors commendably quantify much of their histochemical analysis. Quantification of the data represented in Figure 4F would help to bolster the authors' conclusions of the status of the HIF1-MIF axis in their model. Also, the scale bars are missing in this figure.

3) Related to Figure 5 (manuscript lines 239-244) the authors comment on the relative amount of necrosis in control versus CI-/- tumors. If the authors wish to report on varied levels of necrosis, they should directly measure necrosis or thoroughly describe how their H&E observations support their conclusions on necrosis. Also related to Figure 5C, the circles, arrows, and asterisks are too small to be readily observed.

Reviewer 2 Report

In this study, Kurelac et al. sought to establish a humanized model of the osteosarcoma graft, and examine whether deficiency of mitochondrial complex I (CI) in osteosarcoma cells leads to alterations in tumor microenvironment (TME), in particular bone specific non-neoplastic cells, during cancer progression. The results demonstrated that the new model enabled observation of a mature bone tissue in which osteosarcoma cells progress along with multiple types of mesenchymal cells. Parenchymal cells appeared to be an important component of TME, and abundance of cancer associated fibroblasts and tumor-associated macrophages (TAM) are the hallmarks of TME with CI-deficient osteosarcoma cells (143B-/-). The authors concluded that the pseudo-orthotopic humanized model of osteosarcoma graft allowed analyses of osteosarcoma progression in relatively native microenvironment compared with invasive intra-bone cell transplantation. Furthermore, they claimed that this study identified potential association between inhibition of CI in cancer treatment and preservation of bone tissue. Given that the mitochondrial functions critically affect behavior of tumor cells in a variety of cancers and modulation of mitochondrial functions in cancer has significant potential in clinical medicine, this study seems to be of particular interest to readers of Journal of Clinical Medicine. However, there are several concerns which need to be addressed before publication.

(1) The authors claimed that “the pseudo-orthotopic approach allowed investigation of osteosarcoma progression in its native microenvironment setting”. However, because of the lack of T and B cells in the implanted mice, the model cannot evaluate the influence of anti-tumor effects by TAM associated with Treg recruitment and anti-inflammatory cytokines. Therefore, from this model, it seems impossible to conclude the consequence of CI inhibition in tumor suppression.

(2) In general, the number of samples in quantification is kept very low in this study. This is considered to be critical when the authors claimed that “there is no statistical difference” regarding the size of the tumors and the ratio of Ki-67 positive cells.

(3) The authors claimed that the CI deficiency has no obvious effect on tumor size and Ki-67 index is similar at 60 days after implantation. However, the authors themselves showed that the content of the tumors was distinct, since CI deficient tumors appeared to have much less necrosis, fewer blood vessels, and more connective tissues with TAM and fibroblasts. Therefore, it seems critical to have more detailed time-dependent change of tumor size during progression of tumor. In particular, tumor growth of 143B-/- is likely to be enhanced after 60 days post implantation, which would have significant implication. In addition, Ki67-positive cells should be identified, since Ki-67 needs to be measured in tumor cells rather than non-cancerous cells. Since the authors demonstrate “intensive necrosis” in CI-positive tumors (143+/+), it needs to be clarified how they avoided necrotic regions in histological quantification.

(4) In Fig. 2, the authors claimed that the bone tissues are composed of multiple types of cells of human origin. The authors need to provide evidence showing the observed cells are from human origin.

(5) They quantified macrophages in 3 regions. More detailed information about selections of the regions for quantification to avoid tumor edges and necrotic areas is necessary, given the significant variation among regions.

(6) Authors indicated in Fig. 3 that the percentages of TAM were higher in 143B-/-. However, the tumor sizes were also different between 143+/+ and 143B-/- at day 30 after implantation, and therefore the higher density of macrophages in 143B-/- may be simply due to the condensation of macrophages in smaller tumor tissues rather enhanced infiltration of macrophages due to decreased MIF.

(7) The quality of immunostaining in Fig. 3d is not sufficient to evaluate distribution of macrophages inside the tumors. Along with improving the immunostaining intensity, adding higher mag images and marks on the cells would help.

(8) The list of molecules they evaluated in human cytokine array needs to be clarified.

What about other molecules which potentially attract macrophage infiltration, such as CCL2 and CCL5?

(9) Conditioned media of the large tumor mass does not necessarily the molecules are produced throughout the tumor mass. Is the expression of the MIF changed deep inside the tumor tissues? Is MIF expressed by non-cancerous cells?

(10) The lack of HIF1a and blood vessels detected by endomucin appear to be inconsistent with their notion that “CI deficiency was also associated with the abundance of TAMs, that have been called into play to provide angiogenic factors when cancer cell-autonomous HIF1 signals are absent.”. Are blood vessels less in 143B-/- tumors? How much variance or correlation is observed between HIF1a and blood vessel abundance? Quantification of blood vessel abundance and HIF1a expression in serial sections would help.

(11) The distribution of smooth muscle cell actin (SMA) positive cells is not clear, due to the low quality of the immunostaining. Improving immunostaining intensity, adding higher magnification and adding marks are essential. It is unclear if those SMA-positive cells represent cells associated with blood vessels, such as smooth muscle cells and pericytes.

(12) The number of used animals should be clarified in quantitative comparisons.

(13) Distinguishing osteocyte and osteoblast from other cells nearby on HE sections is not reliable when cell elimination needs to be clarified. Immunostaining for markers of those cells along with nuclear staining will more clearly show the elimination.

(14) Necrosis seems to predominate in the core of the tumors. How about the survival of osteoblasts and osteocytes in 143B+/+ near periphery of the tumors? Does the quantification include those areas, although the counted number of samples and cells appear small?

(15) It is better to provide explaination of “% of Live”.

(16) There is no scale bar in Fig. 4b.

Round 2

Reviewer 2 Report

The authors provided reasonable explanation and appropriate modification to their manuscript. The quality of most images was improved. A few more comments are provided below.

(a1) As the authors admitted, the microenvironment of the model is not "native" setting. Therefore, it would be better to reword "native microenvironment" to phrases such as "bone-like microenvironment which containing transplanted human parenchymal cells".

(a2) In relation to the reviewer’s comment (10), the findings that CI-tumors had higher number but smaller size of blood vessels compared to the control tumors are interesting. However, the intensity of Endomucin is too low to evaluate the abundance of Endo + /FOV.
